# A Pacific Oyster-Derived Antioxidant, DHMBA, Protects Renal Tubular HK-2 Cells against Oxidative Stress via Reduction of Mitochondrial ROS Production and Fragmentation

**DOI:** 10.3390/ijms241210061

**Published:** 2023-06-13

**Authors:** Hsin-Jung Ho, Natsumi Aoki, Yi-Jou Wu, Ming-Chen Gao, Karin Sekine, Toshihiro Sakurai, Hitoshi Chiba, Hideaki Watanabe, Mitsugu Watanabe, Shu-Ping Hui

**Affiliations:** 1Faculty of Health Sciences, Hokkaido University, Sapporo 060-0812, Japan; hsinjung@hs.hokudai.ac.jp (H.-J.H.);; 2Department of Nutrition, Sapporo University of Health Sciences, Sapporo 007-0894, Japan; chibahit@med.hokudai.ac.jp; 3Watanabe Oyster Laboratory, Co., Ltd., Tokyo 192-0154, Japan; 4Graduate School of Science and Engineering, Soka University, Tokyo 192-8577, Japan

**Keywords:** mitochondria, oxidative stress, mitochondrial biogenesis, mitochondrial fusion/fission, mitophagy, mitochondrial respiration

## Abstract

The kidney contains numerous mitochondria in proximal tubular cells that provide energy for tubular secretion and reabsorption. Mitochondrial injury and consequent excessive reactive oxygen species (ROS) production can cause tubular damage and play a major role in the pathogenesis of kidney diseases, including diabetic nephropathy. Accordingly, bioactive compounds that protect the renal tubular mitochondria from ROS are desirable. Here, we aimed to report 3,5-dihydroxy-4-methoxybenzyl alcohol (DHMBA), isolated from the Pacific oyster (*Crassostrea gigas*) as a potentially useful compound. In human renal tubular HK-2 cells, DHMBA significantly mitigated the cytotoxicity induced by the ROS inducer L-buthionine-(S, R)-sulfoximine (BSO). DHMBA reduced the mitochondrial ROS production and subsequently regulated mitochondrial homeostasis, including mitochondrial biogenesis, fusion/fission balance, and mitophagy; DHMBA also enhanced mitochondrial respiration in BSO-treated cells. These findings highlight the potential of DHMBA to protect renal tubular mitochondrial function against oxidative stress.

## 1. Introduction

Mitochondria play an important role in redox signaling systems and energy metabolism by producing reactive oxygen species (ROS) during the utilization of oxygen for ATP synthesis [1]. Mitochondria are exceedingly dynamic organelles that constantly undergo fusion/fission and turnover through mitochondrial biogenesis and mitophagy [2,3]. Mitochondrial fusion reduces stress by joining partially damaged mitochondrial contents as a form of complementation, whereas fission is necessary to clear damaged mitochondria via the segregation of dysfunctional mitochondria [4]. Mitochondrial biogenesis regulates the growth and division of mitochondria, resulting in increased ATP production in cells. However, mitophagy is the selective autophagic process for removing damaged mitochondria [5]. Mitochondria-derived ROS can cause mitochondrial dysfunction in kidney diseases, including chronic kidney disease and diabetic nephropathy [6,7]. In the kidneys, proximal tubular cells are enriched with mitochondria to sustain the substantial aerobic ATP synthesis required for the reabsorption of nutrient-rich glomerular filtrate [8]. Thus, improving mitochondrial dynamics in renal proximal tubular cells is considered a new therapeutic and preventive strategy for kidney diseases.

Recent studies have shown the cytoprotective effects of natural compounds, such as curcumin and quercetin, in reducing oxidative stress in the kidneys. In our previous study, we suggested that these famous natural antioxidants showed higher cytotoxicity, assessed by a correlation between the cytoprotective effects and log *p* values [9]. However, we have isolated a phenolic compound, 3,5-dihydroxy-4-methoxybenzyl alcohol (DHMBA), from the extracts of Pacific oyster (*Crassostrea gigas*). DHMBA was identified as a novel antioxidant with less cytotoxicity and is considered a potentially useful functional food for antioxidant purposes [10]. Furthermore, DHMBA protects hepatocytes from oxidative stress without causing significant cytotoxicity through radical absorption and activation of the Nrf2-dependent antioxidant pathway [11,12]. A computational chemistry study proposed that DHMBA can serve as a metal ion chelator and can be efficiently regenerated in an aqueous solution at physiological pH [13]. In light of the potential antioxidative effects and the safety of DHMBA, we aimed to determine its potential in supporting kidney health. Recognizing the pivotal role of mitochondria in the generation of ROS and the initiation of oxidative stress, we aimed to elucidate the impact of DHMBA on ROS production, as well as the dynamic behavior of mitochondria and mitochondrial biogenesis in renal proximal tubular cells under oxidative conditions. This study suggests that DHMBA protects renal proximal tubular cells from oxidative injury and maintains mitochondrial function.

## 2. Results and Discussion

### 2.1. DHMBA Protected HK-2 Cells from BSO-Induced Oxidative Damage

L-buthionine-(S, R)-sulfoximine (BSO) is one of the most widely used potent inhibitors of γ-glutamylcysteine synthetase. BSO causes glutathione (GSH) depletion and cellular ROS accumulation, leading to cell death [14,15,16]. The effects of BSO and DHMBA on the viability and cytotoxicity of HK-2 cells are shown in Figure 1. Cell viability decreased with 100 μM BSO in the absence of DHMBA but improved in the presence of DHMBA in a dose-dependent manner (2–15.6 μM), remaining at similar levels up to 500 μM (Figure 1a). The cytotoxicity increased up to approximately 4.5-fold with 100 μM BSO, compared with the control in the absence of DHMBA, but it decreased in the presence of DHMBA in a dose-dependent manner (2–15.6 μM), remaining at similar levels up to 500 μM (Figure 1b). Overall, DHMBA effectively protected HK-2 cells from BSO-induced cell death and cytotoxicity.

### 2.2. DHMBA Reduced Mitochondrial ROS Production and Oxidative Mitochondrial Apoptotic Death in BSO-Treated HK-2 Cells

Mitochondrial ROS levels were measured in BSO-treated cells, with and without DHMBA. Mitochondrial ROS production was significantly increased by BSO and decreased by 250 and 500 μM DHMBA (Figure 2a,b). As the mitochondria are the main source of ROS, the observed reduction in mitochondrial ROS levels in DHMBA-treated cells can be explained by the possible protection of mitochondrial function by DHMBA.

The increased expression of BCL2 interacting killer (BIK) and endonuclease G (ENDOG) resulted in mitochondrial-mediated apoptotic cell death [17]. The mRNA expression levels of *BIK* and *ENDOG* were increased by BSO by up to 2.4- and 1.5-fold, respectively, compared to the controls. In the presence of DHMBA, mRNA expression of *BIK* significantly decreased at 250 and 500 μM DHMBA, and mRNA expression of *ENDOG* decreased at 250 μM DHMBA (Figure 2c). Oxidative stress-induced cell death is associated with the activation of mitochondrial apoptotic pathway-related proteins, including BIK and ENDOG. These proteins are considered markers of mitochondrial DNA damage and oxidative stress [17]. BIK is known to act as a pro-apoptotic protein that triggers mitochondrial cytochrome c release and induces apoptosis [18]. ENDOG is a nuclear-encoded mitochondrial-localized nuclease that translocates to the nucleus during apoptosis. While ENDOG is released from the mitochondria, the subsequent increase in cytosolic ENDOG promotes cell death associated with nuclear DNA fragmentation [19,20]. In the present study, DHMBA reduced the mitochondrial ROS and the expression levels of *BIK* and *ENDOG*, supporting the protection of mitochondrial function against oxidative stress by DHMBA.

### 2.3. DHMBA Improved Mitochondrial Morphology in BSO-Treated HK-2 Cells

Oxidative stress causes mitochondrial fragmentation and spherical morphologies, which are associated with apoptosis induction [21] and impaired mitochondrial function. The present study showed that the mitochondria in BSO-induced cells had spherical morphologies. However, 250 and 500 μM DHMBA improved BSO-induced mitochondrial fragmentation (Figure 3a). Similar morphologies were observed in the control and DHMBA-treated cells. Mitochondrial fusion, fission, and biogenesis were identified as playing important roles in mitochondrial morphology and quality control [22]. The pro-fusion protein optic atrophy 1 (OPA1) and pro-fission proteins dynamin-related protein 1 (DRP1) and mitochondrial fission 1 protein (FIS1), are the core proteins that modulate mitochondrial morphology [23]. The mRNA expression level of *OPA1* decreased in BSO-treated cells and tended to increase with 500 μM DHMBA. Moreover, the BSO addition did not change *DRP1* expression with or without DHMBA, whereas *FIS1* expression increased in BSO-treated cells and decreased with 250 μM DHMBA (Figure 3b). Maintaining the balance of mitochondrial fusion and fission improves the mitochondrial respiratory function. Here, mitochondrial fragmentation and fission were enhanced through BSO-induced oxidative stress. However, DHMBA partly regulated the expressions of *OPA1* and *FIS1* to inhibit mitochondrial fission and improve their morphology. Since DRP1 is a cytosolic protein that promotes mitochondrial fission and FIS1 is a transmembrane protein that serves as a receptor for DRP1 and recruits it to the membrane to induce fission [22,23], further investigation is needed to study the interaction of DRP1, FIS1, and other fission protein, such as the mitochondrial fission factor. However, these results indicate that oxidative stress causes an imbalance in mitochondrial fission and fusion, whereas DHMBA improves the balance of mitochondrial dynamics, resulting in a better mitochondrial morphology.

Moreover, the mRNA expression levels of mitochondrial biogenesis-related genes in BSO-treated HK-2 cells were measured (Figure 3c). The PGC1α pathway is involved in mitochondrial biogenesis for the growth and division of pre-existing mitochondria and is considered a master regulator of mitochondrial biogenesis and an attractive therapeutic target. PGC1α regulates genes, including nuclear factor E2-related factor 1 (NRF1), to control the expression of mitochondrial proteins, and mitochondrial transcription factor A (TFAM) to initiate mtDNA transcription and replication [24,25]. The expression of *PGC1α* was decreased in BSO-treated cells. However, DHMBA increased the expression of *PGC1α* and its downstream gene, *TFAM*, but not *NRF1*, in BSO-treated cells. The upregulation of mitochondrial biosynthesis can increase mitochondrial numbers and improve cellular metabolism in response to oxidative stress. PGC1α has also been implicated in metabolic disorders, such as heart failure, diabetes, and renal diseases [26,27,28]. Based on our results, DHMBA increased the expression of *PGC1α* and its target gene, *TFAM*, to enhance mitochondrial biogenesis in cells with oxidative damage. Consequently, DHMBA may maintain mitochondrial function during energy production. The regulation of mitochondrial fusion, fission, and biogenesis by DHMBA suggests that DHMBA enables quality control of the mitochondria against oxidative stress.

### 2.4. DHMBA Inhibited Autophagy/Mitophagy in BSO-Treated HK-2 Cells

The regulation of autophagy and mitophagy is important for preserving cellular integrity and mitochondrial function. Dysregulation of redox signaling can alter autophagic activity, resulting in cell death [29]. Increasing evidence suggest that oxidative stress acts as the main intracellular signal transducer sustaining autophagy [30,31]. Autophagy degrades damaged organelles by engulfing the damaged one with autophagosomes, which are then fused with lysosomes for degradation. LC3 and p62 are autophagy-related proteins that regulate autophagic activity through a complex signaling network [32,33]. LC3 is processed from its cytosolic form (LC3-I) to its lipidated form (LC3-II) during autophagy, and the ratio of LC3-II/I is a commonly used marker of autophagic activity. p62, on the other hand, is a selective autophagy receptor that binds to ubiquitinated cargo and LC3, facilitating the targeting of cargo to the autophagosome for degradation [32]. Therefore, the protein expression ratio of LC3-II/I increased, and the protein expression of p62 decreased in BSO-treated HK-2 cells, indicating enhanced autophagy in oxidative stress cells. DHMBA reduced the LC3-II/I ratio and increased p62 expression (Figure 4a). Consistent with the protein expression, the expression of autophagosome-related genes, *LC3B* and *GABARAPL1*, also increased in BSO-treated HK-2 cells. However, DHMBA decreased the expression of these genes (Figure 4b). p62 binds directly to *LC3* and *GABARAP* family proteins to deliver selective autophagic cargo for degradation. p62 protein is degraded through autophagy and serves as a marker for examining autophagic flux [33]. Thus, DHMBA protected cells from autophagic cell death.

Dysfunctional mitochondria are recognized and degraded through selective autophagy, which is known as mitophagy. We proceeded to determine the expression of related genes to clarify the effects of DHMBA on mitophagy. Dysfunctional mitochondrial elimination can be roughly divided into receptor-mediated and PINK1-mediated mitophagy. Although the expression of receptor-mediated (Figure 4c) and PINK1-mediated (Figure 4d) mRNA increased in BSO-treated cells, DHMBA regulated receptor-mediated mRNA expression rather than PINK1-mediated expression. PINK1 functions as an initiation protein and a signal-amplifying protein in the priming of the mitochondria for mitophagy. Further, receptors on the outer mitochondrial membrane play an important role in mitophagy, which has been described as a major pathway targeting the mitochondria [34]. Two types of mitophagy receptors have been identified in mammalian cells; one family includes BCL2 interacting protein 3 (BNIP3) and BCL2 interacting protein 3 such as (BNIP3L), and the other includes FUN14 domain-containing protein 1 (FUNDC1) [35,36,37]. Modulation of receptor-mediated mitophagy is considered a possible therapeutic approach in several diseases, including cardiovascular diseases [34] and acute kidney injury [38]. Our study revealed that BSO induces autophagic activity through PINK1-mediated and receptor-mediated mitophagy, resulting in cell death. However, DHMBA inhibited mitophagy for mitochondrial breakdown in BSO-treated HK-2 cells, particularly by regulating the receptor-mediated mitophagy.

### 2.5. DHMBA Enhanced Mitochondrial Functions in BSO-Treated HK-2 Cells

Oxidative stress due to the depletion of GSH and overproduction of ROS causes structural damage to proteins, enzymes, DNA, and lipids, leading to various functional loss and cell death [39]. In the present study, mitochondrial breakdown signaling (mitochondrial fragmentation and mitophagy) increased in BSO-treated cells in association with possible mitochondrial dysfunction. These results suggested that DHMBA prevented oxidative stress-induced changes in mitochondrial energy metabolism. Therefore, we measured the mitochondrial oxygen consumption rate (OCR) using an extracellular flux analyzer. A mitochondrial stress assay was used to determine the parameters of mitochondrial oxidative phosphorylation (Figure 5). Basal respiration was first measured under normal conditions. Thereafter, oligomycin was added to inhibit respiratory complex V for ATP-linked respiration, FCCP was added to uncouple the mitochondria for maximal respiration, and rotenone/antimycin A (Rot/AA) was added to inhibit respiratory complexes I and III to stop mitochondrial respiration. The remaining OCR indicated non-mitochondrial respiration in HK-2 cells (Figure 5a,b). The parameters for mitochondrial respiration were calculated according to each OCR. Although all parameters did not change in BSO-treated HK-2 cells for 24 h, the addition of DHMBA increased basal respiration, ATP-linked respiration, and maximal respiration (Figure 5c). The ROS generated mainly in the respiratory chain and higher ROS levels cause mitochondrial permeability transition pore to be open to destroy mitochondria, leading to cell and organismal death [7]. The addition of BSO increased mitochondrial ROS accumulation in a 24-h period (Figure 2a) without changed mitochondrial respiration. However, DHMBA enhanced mitochondrial respiration in the BSO-treated cells. Thus, DHMBA might enhance mitochondrial respiration to maintain cellular homeostasis before the mitochondria are destroyed by oxidative stress. Collectively, DHMBA may be useful for mitochondrial quality control in oxidative stress-related diseases.

## 3. Materials and Methods

### 3.1. Materials

The DHMBA was synthesized as previously described [10,40]. The L-Buthionine-(S, R)-sulfoximine (BSO) was purchased from Sigma-Aldrich (St. Louis, MO, USA). The dihydrorhodamine 123 (DHR123) was purchased from FUJIFILM Wako Pure Chemical Corp. (Tokyo, Japan).

### 3.2. Cell Line and Culture Conditions

The human renal proximal tubular HK-2 cells were obtained from the American Type Culture Collection (ATCC, Manassas, VA, USA). The cells were maintained in a low-glucose Dulbecco’s modified Eagle’s medium (DMEM, Nacalai Tesque, Kyoto, Japan) containing 10% fetal bovine serum (FBS, Biosera, East Sussex, UK), 50 U/mL penicillin, and 50 mg/mL streptomycin under 5% CO_2_ at 37 °C.

### 3.3. Cell Viability and Cytotoxicity

The HK-2 cells were seeded into 96-well plates (4.0 × 10^3^ cells/well) and cultured overnight. Thereafter, the cells were exposed to DHMBA (2–500 μM) and 100 μM BSO for 48 h. The cell toxicity of BSO and cell protective effects of DHMBA in HK-2 cells were measured using the Cell Counting Kit 8 (CCK-8 assay, Dojindo Laboratories, Kumamoto, Japan) and the LDH Cytotoxicity Detection Kit (LDH assay, TaKaRa Shuzo, Shiga, Japan) to assess the cell viability and cytotoxicity, respectively. All the procedures were performed according to the manufacturer’s instructions.

### 3.4. Mitochondrial ROS Levels

The HK-2 cells were seeded into 96-well plates (4.0 × 10^3^ cells/well) and cultured overnight. The cells were then exposed to DHMBA (250 and 500 μM) and 100 μM BSO for 24 h. Mitochondrial ROS production was measured using the fluorogenic probe DHR123, which is a membrane-permeable dye that is oxidized to rhodamine and subsequently localized to the mitochondria [41]. The cells were incubated with 5 μM DHR123 in FBS-free DMEM for 40 min at 37 °C and then washed twice with D-PBS (-). The fluorescence intensity of DHR123 (Ex/Em = 485/535 nm) was determined using a Wallac 1420 ARVO Mx plate reader (PerkinElmer, Tokyo, Japan).

### 3.5. Microscopic Imaging

The HK-2 cells were seeded into 35-mm dishes (0.5 × 10^5^ cells/well) and incubated overnight. Thereafter, the cells were exposed to DHMBA (250 and 500 μM) and 100 μM BSO for 24 h. The mitochondria and nuclei of the HK-2 cells were stained with 0.1 μM MitoTracker Green (Invitrogen, Carlsbad, CA, USA) and 5 μg/mL Hoechst33342 (Dojindo) for 20 min at 37 °C. After washing twice with D-PBS(-), images were acquired using a BZ-9000 All-in-one Fluorescence Microscope (KEYENCE, Osaka, Japan).

### 3.6. mRNA Expression 

The HK-2 cells were seeded into 6-well plates (0.5 × 10^5^ cells/well) and then exposed to DHMBA (250 and 500 μM) and 100 μM BSO for 24 h. Total RNA was isolated using a NucleoSpin kit (TaKaRa Shuzo), and the RNA quality was determined using a Nanodrop One spectrophotometer (Thermo Fisher Scientific, Waltham, MA, USA). ReverTra Ace^®^ qPCR RT Master Mix with gDNA remover (TOYOBO, Osaka, Japan) was used for cDNA synthesis. All the procedures were performed according to the manufacturer’s instructions. mRNA expression levels were determined through a quantitative reverse transcription-polymerase chain reaction (qRT-PCR). The qRT-PCR mixture comprised 2 μL cDNA, 5 μL THUNDERBIRD^®^ SYBR^®^ qPCR Mix (TOYOBO), 0.5 μL of each primer (6 μM), and RNase-free water added to a final volume of 10 μL. The reaction was performed according to the manufacturer’s protocol. Single amplicon amplification was confirmed using melting curve analysis. qRT-PCR was conducted using the CFX Connect Real-Time PCR Detection System (Bio-Rad). β-actin was used as an endogenous control to normalize the results for each sample using the comparative ΔΔCt (cycle threshold) method. Oligonucleotides were obtained from FASMAC (Kanagawa, Japan), and the list of primers used is provided in Appendix A.

### 3.7. Protein Expression

The total protein was analyzed using sodium dodecyl sulphate-polyacrylamide gel electrophoresis (SDS-PAGE) and western blotting. Briefly, the HK-2 cells were seeded into 6-well plates (0.5 × 10^5^ cells/well) and incubated overnight. The following day, the cells were exposed to DHMBA (250 and 500 μM) and 100 μM BSO for 24 h. Cell lysates were prepared using radio-immunoprecipitation assay buffer (Nacalai Tesque) according to the manufacturer’s protocol. Protein concentrations were measured using a Pierce™ BCA Protein Assay Kit (Thermo Fisher Scientific, Waltham, MA, USA). Lysates were mixed with equal volumes of EzApply (Atto, Tokyo, Japan) and heated at 95 °C for 5 min. For SDS-PAGE, equal amounts of protein (10 μg) were separated on a precast polyacrylamide gel (5–20% e-PAGE, Atto) at a constant voltage of 100 V for 100 min. Proteins were then transferred onto a 0.2 μm polyvinylidene difluoride membrane (Millipore, Bedford, MA, USA). The membrane was blocked against non-specific reactions with 0.5% (*w*/*v*) skim milk and diluted in Tris-buffered saline containing 0.1% (*v*/*v*) Tween 20 (TBS-T). Thereafter, the membranes were incubated overnight at 4 °C with each antibody, washed with TBS-T, incubated with horseradish peroxidase-conjugated secondary antibodies, and washed again. The stained bands were visualized using the KPL LumiGLO^®^ chemiluminescent substrate system (Seracare Life Sciences, Milford, MA, USA). Images were captured and documented using a CCD system (ChemiDocTM MP Imaging System, Bio-Rad, Hercules, CA, USA). The relative expression level of each protein was normalized to that of β-actin. The primary antibodies used in this study were purchased from MBL Co., Ltd. (Nagoya, Japan), including p62 (1:1000, M162-3), LC3 (1:1000, M186-3), and β-actin (1:1000, M177-3).

### 3.8. Measurement of Mitochondrial Oxygen Consumption Rate (OCR)

To assess mitochondrial function, the OCR and pH gradient of HK-2 with DHMBA were measured using a seahorse XFp analyzer (Agilent Technologies, Santa Clara, CA, USA). Briefly, HK-2 cells were seeded in a mini-plate (1.0 × 10^4^ cells), incubated overnight, and treated with 100 μM BSO and 500 μM DHMBA in XF DMEM (Agilent Technologies) for another 24 h. For equilibration, the mini-plates were cultured in a XF Calibrant Solution assay medium without CO_2_ for 60 min. After equilibration, the cells were measured, and then injections of the Mito Stress Test Kit (Agilent Technologies), including oligomycin (respiratory complex V inhibitor, 2 µM), carbonyl cyanide-p-trifluoromethoxyphenylhydrazone (FCCP, uncoupler, 1 µM), and rotenone/antimycin a (Rot/AA, inhibitors of respiratory complex I and III, 0.25 µM), were performed for mitochondrial oxidative phosphorylation. Multiple parameters were obtained in this assay, including basal respiration, ATP-linked respiration, maximal and reserve capacities, and non-mitochondrial respiration.

### 3.9. Statistical Analysis

The data are expressed as mean ± standard deviation (SD). The comparisons were performed using a one-way analysis of variance followed by a Dunnett’s multiple comparisons test (JMP^®^Pro16, SAS Institute, Cary, NC, USA). Differences were considered statistically significant at *p* < 0.05.

## 4. Conclusions

Increasing mitochondrial ROS levels caused mitochondrial breakdown in HK-2 cells treated with BSO. However, DHMBA ameliorated mitochondrial damage and reduced the cytotoxicity of excessive ROS. Such findings highlight the potential of DHMBA in protecting mitochondrial function in renal tubular cells.

There are limitations that should be acknowledged. The experiments were conducted in a cell-based model and may not cover all the aspects of mitochondrial function and dynamics, which may not fully reflect the changes of kidney physiology. Despite these limitations, our study highlights the potential of DHMBA as a mitochondrial protective agent for reducing oxidative stress and maintaining better proximal tubular function. Further studies are needed to confirm the efficacy of DHMBA in animal models and human clinical trials.

## 5. Patents

A Japan patent was filed resulting from the work reported in this manuscript: Patent Application No. 022-53125.

## Figures and Tables

**Figure 1 ijms-24-10061-f001:**
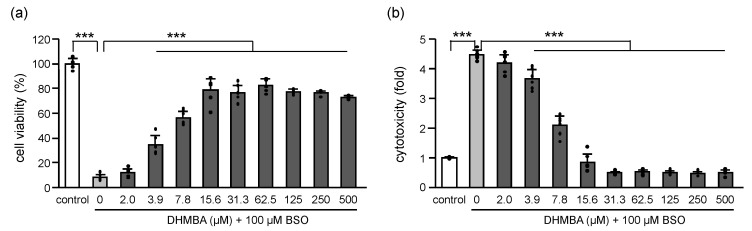
Effects of DHMBA on cell viability and cytotoxicity in BSO-treated HK-2 cells. (**a**) Cell viability and (**b**) cytotoxicity were measured using the CCK-8 and LDH assays, respectively. Values are presented as means ± SD (*n* = 5–6). *** *p* < 0.001 vs. 100 μM BSO without DHMBA using the Dunnett’s test.

**Figure 2 ijms-24-10061-f002:**
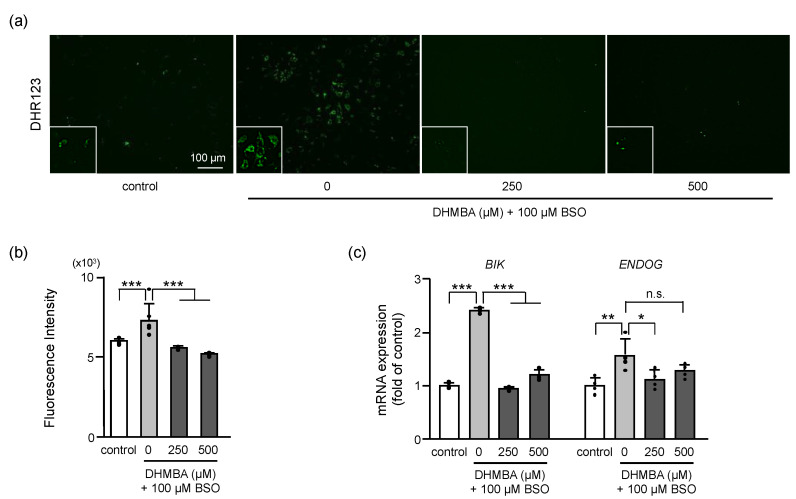
Effects of DHMBA on mitochondrial ROS production and damage in BSO-treated HK-2 cells. (**a**) Mitochondrial ROS production was measured using the fluorogenic probe DHR123. Images of DHR123 staining following treatment with 0, 250, 500 μM DHMBA with or without 100 μM BSO and (**b**) its quantitative presentation. Values are presented as means ± SD (*n* = 6). (**c**) The mRNA expression levels of BIK and ENDOG based on qRT-PCR. Values are presented as means ± SD (*n* = 4). * *p* < 0.05, ** *p* < 0.01, *** *p* < 0.001 vs. 100 μM BSO without DHMBA using the Dunnett’s test. n.s.: not significant.

**Figure 3 ijms-24-10061-f003:**
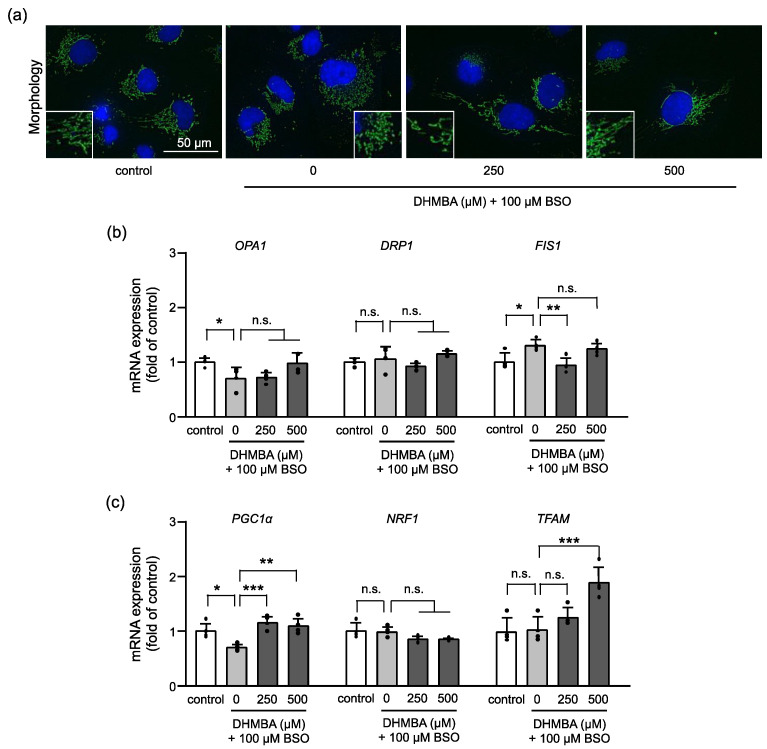
Effects of DHMBA on mitochondrial morphology in BSO-treated HK-2 cells. (**a**) Images of the mitochondria and nuclei using MitoTracker green and Hoechst33342, respectively. (**b**) The mRNA expression levels of OPA1, DRP1, and FIS1 based on qRT-PCR. β-actin was used as an internal control. (**c**) The mRNA expression of the mitochondrial biogenesis-related gene based on qRT-PCR. β-actin was used as an internal control. The data are expressed as the fold induction relative to the control. Values are presented as means ± SD (*n* = 4). * *p* < 0.05, ** *p* < 0.01, *** *p* < 0.001 vs. 100 μM BSO without DHMBA by the Dunnett’s test. n.s.: not significant.

**Figure 4 ijms-24-10061-f004:**
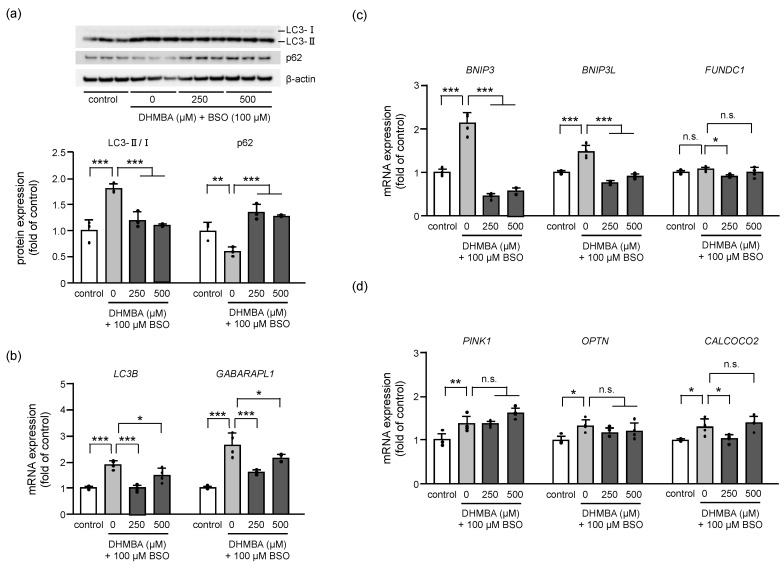
Effects of DHMBA on mitophagy in BSO-treated HK-2 cells. (**a**) The expression levels and quantitative data for autophagy-related protein based on western blot analysis. The protein levels were normalized to that of β-actin. Values are presented as means ± SD (*n* = 3). (**b**) The expression of autophagy-related mRNA based on qRT-PCR. The level of β-actin expression was used as an internal control. Values are presented as means ± SD (*n* = 4). (**c**) The expression of mRNA for receptor-mediated mitophagy. (**d**) The expression of mRNA for PINK1-mediated mitophagy. β-actin was used as an internal control. The data are expressed as fold induction relative to the control. Values are presented as means ± SD (*n* = 4). * *p* < 0.05, ** *p* < 0.01, *** *p* < 0.001 vs. 100 μM BSO without DHMBA by the Dunnett’s test. n.s.: not significant.

**Figure 5 ijms-24-10061-f005:**
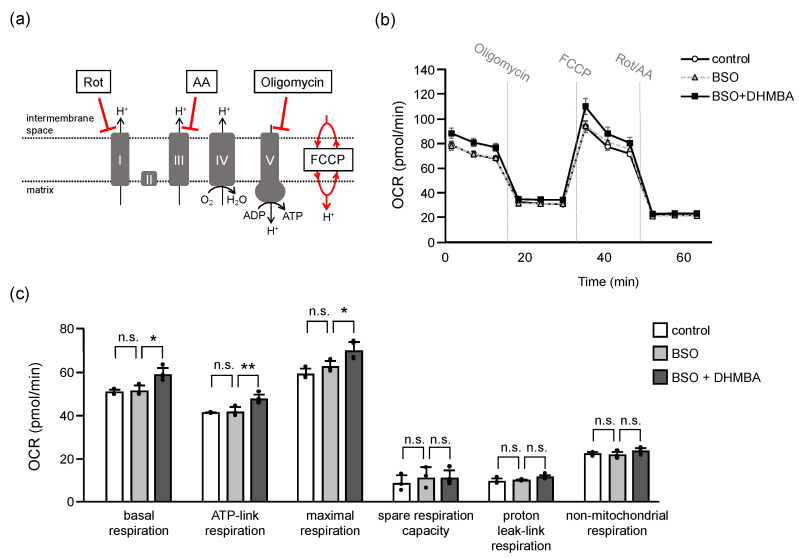
Effects of DHMBA on mitochondrial function in BSO-treated HK-2 cells. (**a**) Schematic diagram of respiratory complex inhibition. (**b**) Mitochondrial respiratory profile. OCR was measured over time (pmol/min) using an extracellular flux analyzer. The cells were measured and then injections of the Mito Stress Test Kit were performed for mitochondrial oxidative phosphorylation. (**c**) The parameters were calculated according to each OCR of mitochondrial oxidative phosphorylation, including basal respiration, ATP-link respiration, maximal respiration, spare respiration capacity, proton leak-link respiration, and non-mitochondrial oxygen respiration. Values are presented as means ± SD (*n* = 3–4). * *p* < 0.05, ** *p* < 0.01 vs. 100 μM BSO without DHMBA using the Dunnett’s test. n.s.: not significant.

## Data Availability

All the Data Availability Statements are included in the paper.

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
