# Peer review of "A Pacific Oyster-Derived Antioxidant, DHMBA, Protects Renal Tubular HK-2 Cells against Oxidative Stress via Reduction of Mitochondrial ROS Production and Fragmentation"

_ijms, 2023, doi:10.3390/ijms241210061_

Round 1

Reviewer 1 Report

Critique

1.      The major concern about this manuscript is that all the studies have been done just with one cell line. The authors should try to reproduce at least some of the critical findings in HK-2 cells with some other renal proximal tubular cells (primary cells or immortalized cell lines).

2.      How does DHMBA reduce ROS? We see a decrease in ROS, but how does the compound elicit this effect? The data show that DHMBA reverses the effect of BSO, but has this anything to do with cellular glutathione levels? Does DHMBA induce NRF2 and hence glutathione synthesis by overcoming the inhibitory actions of BSO? The expression levels of the two enzymes involved in GSH biosynthesis need to be analyzed and cellular glutathione levels measured.

3.      It is stated that DHMBA chelate iron. Does this compound chelate iron? Does it increase lipid peroxidation and increase the production of malondialdehyde? If it does, could this be an underlying mechanism (ie., prevention of the Fenton reaction) for the antioxidant features of DHMBA.

4.      There is no information anywhere in the manuscript as to why DHMBA targets only mitochondria. Does the compound enter mitochondria?

5.      How stable is the compound in HK-2 cells? Apparently, it enters the cells, but what happened to the methoxy group at position? Does this group get changed to hydroxyl group to generate 3,4,5-trihydroxy benzene alcohol? Otherwise, ti is difficult to see how this compound can bind ions. 

6.      Change the title to the present tense: “protects” instead of “protected”.

7.      Page 2, Results section: Do the authors mean 2-15.6 uM instead of 2-5.6 uM for the dose-dependent protection of the cells by DHMBA against BSO-induced cell death? For cytotoxicity, it seems that the range should be 2-31.3 uM instead of 2-15.6 uM.

8.      Page 3, 2nd para: It is not overexpression of ENDOG in the cytoplasm if it comes from the mitochondrial release. It is an increase in the levels of ENDOG in the cytoplasm.

Author Response

Thank you for your insightful critique. In response to the comments and concerns raised, we have prepared a detailed point-by-point response. Please see the attachment.

Reviewer 2 Report

In this study the authors investigated how a pacific oyster-derived antioxidant DHMBA, protected renal tubular HK-2 cells against oxidative stress via reduction of mitochondrial ROS production and fragmentation. The study methods and experimental layout were carefully done. Result supports claim and findings have potential for high impact to the field. I have minor concerns to the manuscript.

1.      Any reason why the basal rate of BSO + DHMBA was high?

2.      What is the effect of DHMBA alone on OCR?

No problem with English language

Author Response

Thank you for providing us with your invaluable feedback on our work. In response to the comments and concerns raised, we have prepared a detailed point-by-point response. Please see the attachment.
